# Cytokinesis D is Mediated by Cortical Flow of Dividing Cells Instead of Chemotaxis

**DOI:** 10.3390/cells8050473

**Published:** 2019-05-17

**Authors:** Yuki Tanaka, Md. Golam Sarowar Jahan, Tomo Kondo, Masaki Nakano, Shigehiko Yumura

**Affiliations:** 1Graduate School of Sciences and Technology for Innovation, Yamaguchi University, Yamaguchi 753-8511, Japan; frog7.atyu-bnai@ezweb.ne.jp (Y.T.); sjahan.biochem@ru.ac.bd (M.G.S.J.); tomokondo@bio.c.u-tokyo.ac.jp (T.K.); i023vc@yamaguchi-u.ac.jp (M.N.); 2Department of Biochemistry and Molecular Biology, Faculty of Science, University of Rajshahi, Rajshahi 6205, Bangladesh; 3Department of Life Sciences, Graduate School of Arts and Sciences, The University of Tokyo, Tokyo 153-8902, Japan

**Keywords:** chemotaxis, cortical flow, cytokinesis, *Dictyostelium*, midwife

## Abstract

Cytokinesis D is known as the midwife mechanism in which neighboring cells facilitate cell division by crossing the cleavage furrow of dividing cells. Cytokinesis D is thought to be mediated by chemotaxis, where midwife cells migrate toward dividing cells by sensing an unknown chemoattractant secreted from the cleavage furrow. In this study, to validate this chemotaxis model, we aspirated the fluid from the vicinity of the cleavage furrow of a dividing *Dictyostelium* cell and discharged it onto a neighboring cell using a microcapillary. However, the neighboring cells did not show any chemotaxis toward the fluid. In addition, the cells did not manifest an increase in the levels of intracellular Ca^2+^, cAMP, or cGMP, which are expected to rise in chemotaxing cells. From several lines of our experiments, including these findings, we concluded that chemotaxis does not contribute to cytokinesis D. As an alternative, we propose a cortical-flow model, where a migrating cell attaches to a dividing cell by chance and is guided toward the furrow by the cortical flow on the dividing cell, and then physically assists the separation of the daughter cells.

## 1. Introduction

Cytokinesis is the final step of cell division. *Dictyostelium* cells have four modes of cytokinesis—cytokinesis A, B, C, and D [1]—although recent studies revised this categorization [2]. Cytokinesis D was observed for the first time in Amoebozoa, *Entamoeba invadens* [3]. In that study, neighboring cells migrated toward dividing cells and cut the connection between two daughter cells. When the fluid from the vicinity of the cleavage furrow of a dividing *Entamoeba* cell was aspirated with a micropipette, and then discharged onto distant cells, 37% of the observed cells extended a directed pseudopod and followed a retracting pipette [3]. Therefore, Biron et al. [3] proposed that the neighboring cells are guided by a chemoattractant secreted by dividing cells and facilitate cytokinesis as a midwife. Additionally, in *Dictyostelium* cells, neighboring cells often migrate toward dividing cells and cross the cleavage furrow [4,5]. Nagasaki and Uyeda [6] have observed that the green fluorescent protein (GFP)-tagged pleckstrin homology (PH) domain localizes at the leading edge of midwife cells migrating toward the dividing cell. Since the GFP–PH domain localizes at the leading edge of chemotaxing cells in the aggregation stage of this organism, the authors assumed that midwife cells migrate toward the dividing cell because the midwife cells sense the chemoattractant secreted by the dividing cell. They refer to it as cytokinesis D to distinguish this phenomenon from the other cytokinesis modes [1]. *Entamoeba* and *Dictyostelium* are phylogenetically widely separated. Thus, cytokinesis D may be common among diverse groups of animal and amoeboid cells. Nonetheless, the chemoattractant and signal mechanism, including its receptor, remain unknown.

In this study, we reassessed the chemotaxis model for cytokinesis D. According to the findings made in this study, we concluded that midwife cells do not migrate chemotactically. We propose a novel model, namely, a cortical-flow model, in which migrating cells accidentally attach to dividing cells. They are guided toward the furrow by the cortical flow on the dividing cell and then cross the cleavage furrow, which facilitates the separation of daughter cells. 

## 2. Materials and Methods

### 2.1. Cell Culture

*Dictyostelium discoideum* cells (AX2) were cultured in plastic dishes at 22 °C in the HL5 medium (1.3% of bacteriological peptone, 0.75% of yeast extract, 85.5 mM d-glucose, 3.5 mM Na_2_HPO_4_⋅12H_2_O, and 3.5 mM KH_2_PO_4_, pH 6.3), as described previously [7]. The cells were transformed with extrachromosomal vectors for the expression of the GFP–PH domain, GFP–lifeact, Flamindo2, Dd-GCaMP6s, or Dd-Green cGull by electroporation or laser-poration, as described elsewhere [8,9]. Dd-Green cGull served as a cGMP_i_ probe, in which the codon usage of the original Green cGull [10] was optimized for *Dictyostelium*. The transformed cells were selected in the HL5 medium supplemented with 10 µg/mL G418 (Wako, Osaka, Japan). To observe cytokinesis D, cells were incubated in the HL5 medium. To obtain developed cells in the aggregation stage, after the medium was replaced with BSS (3 mM CaCl_2_, 10 mM KCl, 10 mM NaCl, 3 mM MES, pH 6.3), the cells were maintained at 22 °C after incubation at 10.5 °C overnight. 

### 2.2. Microscopy

Cells were placed in a glass bottom dish and examined under an inverted differential-interference contrast (DIC) microscope or a phase contrast microscope (IX71, Olympus, Tokyo, Japan). The cells expressing the GFP–PH domain, GFP–lifeact, Flamindo2, Dd-GCaMP6s, or Dd-Green cGull were examined under a confocal microscope (LSM510, Zeiss, Oberkochen, Germany) equipped with a 60× objective and an argon laser (with standard filter settings for GFP). Traction force microscopy was conducted as previously described [11]. For interference reflection microscopy (IRM), the excitation filter (555/25 nm for tetramethyl rhodamine, TRITC), the beam splitter (standard four colors), and the emission filter (525/50 nm for GFP) were employed. IRM visualizes the cell-substratum adhesion as dark areas [12]. Time-lapse images were acquired at an interval of 5 or 10 s. To prepare agar-overlaid cells, a 1.5% agarose block was overlaid on the cells according to a previously published method [13]. Microcapillaries were made from glass tubes (GD-1, Narishige, Tokyo, Japan) by means of a puller (PB-7, Narishige). The microcapillary was set on a micromanipulator (MO-3, S, Tokyo, Japan), which was installed with the inverted microscope. For experiments involving polystyrene beads, carboxylated beads (0.5, 1.0, 3.0, or 6.4 µm in diameter, Polysciences, Warrington, PA, USA) were added to the cells in the glass bottom dish. To fix the cells, they were incubated with 2.5% formalin in BSS for 30 min and washed with BSS more than five times.

### 2.3. Microfluidic Experiments

A flow chamber (µ-Slide I Luer, channel volume 150 µl, Ibidi, Munchen, Germany) was connected with a reservoir containing the HL5 medium and a peristaltic pump. The shear stress was set to 0.06 dyn/cm^2^, which did not affect cell migration. In chemotaxis experiments, BSS was used in place of the HL5 medium.

### 2.4. Image Analysis

The acquired images were analyzed in the ImageJ software (http://rsbweb.nih.gov/ij/). For cell tracking, a plugin allowing for “manual tracking” in the ImageJ software was used. Cell velocity was calculated from the centroid after the cell outline was obtained. Time course data on the fluorescence intensities of Flamindo2, Dd-GCaMP6s, and Dd-Green cGull in chemotactic and midwife cells were analyzed with the same software.

To normalize the cell division stages, the mitosis stage index (MSI) was used, as previously described [11]. The MSI was calculated from the long axis (L) and short axis (l) of the cell. The short axis represents the width of the cleavage furrow. The MSI is calculated using the following formula. 

MSI = (L − l)/L(1)

When the MSI is 0, the cell shape is round. When the MSI is 1.0, cell division is complete. The duration of cytokinesis (cytokinesis time) was defined as the time from the initiation of furrowing to the final separation. Graphs were created in GraphPad Prism 7 (GraphPad Software Inc., San Diego, CA, USA) on the basis of the calculations performed in Microsoft Excel.

### 2.5. Statistical Analysis

Statistical analysis was performed using the GraphPad Prism 7. Data are presented as the mean ± standard deviation (SD) and were analyzed by a Student’s *t* test for a comparison between two groups or by one-way ANOVA with Tukey’s multiple-comparison test.

## 3. Results and Discussion

### 3.1. Neighboring Cells Facilitate Cell Division

When *Dictyostelium* cells enter the mitotic phase, they cease migration, assume a round shape, elongate, and constrict the cleavage furrow to separate into two daughter cells. Neighboring cells often migrate toward dividing cells and cross the cleavage furrow. Figure 1A shows a representative time-lapse image of cytokinesis D (Appendix A). The cells were mildly compressed under the agar overlay to improve the image quality. Figure 1B shows a schema of cytokinesis D. Frequencies of cytokinesis D depend on the cell density and were found to be 4.12% ± 0.95% at a cell density of approximately 1,500 cells/mm^2^, 2.79% ± 0.69% at a cell density of ~750 cells/mm^2^, and 1.78% ± 0.76% at a cell density of ~300 cells/mm^2^ (n ≥ 1500 dividing cells in each of the three experiments).

If the neighboring cells facilitate cytokinesis as midwife cells, then the time required for cytokinesis must be reduced by this process. Figure 1C shows that cytokinesis time was significantly reduced by midwife cells (183 ± 32 s [mean ± SD] during normal cell division, 143 ± 27 s with midwife cells, n = 105 and 57, respectively). Figure 1D shows representative time course data from phase contrast microscopy (Phase) and IRM of dividing cells without (Normal) cells and with midwife cells (Midwife), respectively. The IRM visualizes cell–substratum adhesions as dark areas. In all cases (n = 40 cells), the midwife cells penetrated the narrow space between the cleavage furrow and the substratum. Presumably, this penetration by the midwife cell physically facilitates the separation of the daughter cells.

### 3.2. Verification of the Chemotaxis Model (1)

According to the chemotaxis model, an unknown chemotactic substance is secreted from the cleavage furrow of a dividing cell, and neighboring cells migrate toward the furrow because they sense the substance. Biron et al. [3] aspirated fluid from the vicinity of the cleavage furrow of a dividing *Entamoeba* cell with a micropipette and discharged it in the proximity of a distant cell. They reported that 37% of the examined cells showed extension of directed pseudopods and followed a retracting pipette.

In the present study, similar experiments were conducted on *Dictyostelium* cells. Figure 2A depicts a representative experiment. After fluid from the vicinity of the cleavage furrow of a dividing cell was aspirated with a micropipette (upper images in Figure 2A) and discharged near the interphase cells (lower images in Figure 2A), the interphase cells neither extended a directed pseudopod nor followed the micropipette. We obtained the same results in multiple experiments (n = 51). As a control, when cAMP, which is a chemoattractant for developed *Dictyostelium* cells, was applied to the developed cells, they migrated toward the micropipette (Figure 2B). Incidentally, we applied folic acid, which is a chemoattractant for vegetative cells, to vegetative AX2 cells, but the axenic AX2 strain used in this study showed little or no response. This result is consistent with another report [14]. Therefore, we applied cAMP to developed cells as a control.

Nagasaki and Uyeda [6] have provided support to the chemotaxis model in relation to *Dictyostelium* cells adducing that the GFP–pleckstrin homology (PH) domain of protein kinase B, which is a marker of phosphatidylinositol (3,4,5) trisphosphate, localizes at the leading edge of midwife cells migrating toward a dividing cell. Because the GFP–PH domain localizes at the leading edge of a chemotaxing cell [15], these authors concluded that midwife cells chemotactically migrate toward the dividing cell. In the present study, we confirmed that the GFP–PH domain localizes at the leading edge of midwife cells (arrowheads in Figure 2C, Appendix A). Nevertheless, the GFP–PH domain also localized at the leading edge of migrating non-midwife cells in the absence of chemotaxis (Figure 2D), which is in agreement with more recent observations [14]. 

Nagasaki and Uyeda [6] have reported that mutant cells deficient in the G protein β subunit, which mediates *Dictyostelium* chemotaxis via surface receptors, show a lower frequency of cytokinesis D than wild-type cells. We can explain this finding as follows. Most mutant cells cannot encounter dividing cells in time because the average velocity of mutant cells (1.96 ± 0.57 µm/min, n = 60) is much lower than that of wild-type cells (5.67 ± 1.28 µm/min, n = 60). 

### 3.3. Verification of the Chemotaxis Model (2)

The above micropipette experiments indicate that midwife cells do not migrate chemotactically. However, it is possible that the concentration of the putative unknown chemotactic substance is not sufficient to attract cells in the micropipette experiments. Therefore, we next examined the intracellular concentrations of cAMP (cAMP_i_), Ca ions (Ca_i_^2+^), and cGMP (cGMP_i_) in midwife cells. It is known that when cAMP, which is a chemoattractant for *Dictyostelium*, is applied to developed cells, the levels of cAMP_i_, Ca_i_^2+^, and cGMP_i_ transiently increase under the action of cell surface receptors [16,17,18,19]. To test whether the levels of cAMP_i_, Ca_i_^2+^, and cGMP_i_ increase in midwife cells, a cAMP sensor (Flamindo2), a Ca_i_^2+^ sensor (Dd-GCaMP6s), and a cGMP sensor (Dd-Green cGull) were expressed in the cells, respectively. 

When cAMP was applied as a control to the developed cells, cAMP_i_ transiently increased because the fluorescence due to Flamindo2 transiently decreased (Figure 3A, Flamindo2, Chemotaxis). However, midwife cells did not show any notable increase in cAMP_i_ concentration (Figure 3A, Flamindo2, Midwife, Appendix A). Additionally, Ca_i_^2+^ levels transiently increased (Figure 3A, GCaMP, Chemotaxis). However, midwife cells did not show any increase in Ca_i_^2+^ concentration (Figure 3A, GCaMP, Midwife, Appendix A). Furthermore, although cGMP_i_ levels transiently increased (Figure 3A, cGull, Chemotaxis), midwife cells did not show any increase in cGMP_i_ concentration (Figure 3A, cGull, Midwife, Appendix A).

Figure 3B–D show representative time course data on each reaction of cAMP_i_, Ca_i_^2+^, and cGMP_i_ in chemotactic (red) and midwife (black) cells, respectively. Incidentally, the dividing cells did not show any increases in cAMP_i_, Ca_i_^2+^, and cGMP_i_ levels either. Similar results were confirmed in multiple cells (n > 30) in each case.

Collectively, these data mean that the midwife cells do not undergo any changes in the levels of cAMP_i_, Ca_i_^2+^, or cGMP_i_, which suggests that they do not migrate *chemotactically* toward the dividing cells. On the other hand, the putative chemotactic substance may not necessarily affect cAMP_i_, Ca_i_^2+^, and cGMP_i_.

### 3.4. Verification of the Chemotaxis Model (3)

Next, cytokinesis D was examined in a microfluidic chamber. If the medium constantly flows in one direction, the hypothetical unknown substance secreted by dividing cells will flow away and, therefore, only the midwife cells down the stream will migrate toward the dividing cell (Figure 4A). It is known that cells can migrate against shear flow [20,21]. To prevent this reaction, we first optimized the flow rate and set the shear stress to 0.06 dyn/cm^2^. Figure 4B illustrates the results of tracking migrating cells without flow (upper panel) and with flow (lower panel). The optimized flow rate did not affect cell migration (n = 60 cells, each).

Next, the chemotaxis index (CI) was compared between the upstream and downstream locations within a circle with a 100-µm diameter (yellow circle), while a dividing cell was located at the center of the circle (Figure 4C). In this case, the CI was defined as the cosine of the angle consisting of the following three points: the initial position (red), the position of the furrow of a dividing cell (yellow), and the displaced position (green) after 3 minutes of cell migration (Figure 4D). If the cell migrates straight toward the dividing cell, then the CI takes the value of 1.0, and if the cell recedes straight away, then the CI is −1.0. Figure 4E,F show plots of the CI versus the distance of a cell from the dividing cell with and without flow (n = 212 and 215 cells, three independent experiments). There was no significant difference between the average CI without flow (0.066) and with flow (0.036). In the control, when developed cells chemotactically migrated toward the aggregation center, the average CI was 0.909 (n = 157) and was calculated from the angle involving the aggregation center in place of the position of the dividing cell (Figure 4G). 

Collectively, these findings reveal that neighboring cells are not directed by dividing cells. Thus, we concluded that midwife cells are not affected by chemotaxis toward the dividing cells.

### 3.5. The Cortical-Flow Model

When a migrating cell attaches to a dividing cell, the following three events are observed (Figure 5A). (1) The cell migrates along the surface of the dividing cell toward the cleavage furrow. (2) The cell migrates along the surface of the dividing cell toward the pole. (3) The cell departs from the dividing cell. Figure 5B presents the frequencies of these three events. Of note, the cells more frequently migrated along the surface of the dividing cell toward its cleavage furrow (70 ± 8%, n = 186).

Midwife cells may exert force to separate the daughter cells. The traction force was examined during cytokinesis D by traction force microscopy [11]. Figure 5C shows a representative DIC image, a fluorescence image of GFP–lifeact (a marker of actin filaments), a traction map, and a vector map during cytokinesis D. The traction force of the dividing cell was exerted toward the furrow from both daughter cells. When the midwife cell passed under the cleavage furrow, the traction force of dividing cells significantly diminished (the arrow in Figure 5D), which indicates that the insertional migration of the midwife cell cut the connection between daughter cells. This results in the relaxation of the traction force of the dividing cell.

It has been previously known that the cortical actin cytoskeleton flows toward the cleavage furrow in dividing cells [7,22]. As an alternative model of cytokinesis D, we hypothesized the following: a neighboring cell freely migrates, attaches to a dividing cell by chance, and is guided toward the cleavage furrow by the cortical flow. To assess this cortical-flow model, carboxylated polystyrene beads of various sizes were attached to the surface of dividing cells. The attached beads always moved toward the cleavage furrow (Figure 6A, Appendix A). Next, chemically fixed cells (i.e., dead cells) were attached to the surface of the dividing cells. These cells also moved toward the cleavage furrow (Figure 6A, Appendix A).

To evaluate the efficiency of cytokinesis with attached beads or fixed cells, cytokinesis time was examined in each case (Figure 6B). Both larger beads and fixed cells increased cytokinesis time. Because smaller beads (0.5 and 1 µm) did not affect cytokinesis time, the observed retardation may be due to their heavy load. Since cytokinesis time was significantly shorter in cytokinesis D with live cells (Midwife in Figure 6B), live cells are, therefore, required for efficient cytokinesis D. 

For the cortical-flow model, cell–cell adhesion is essential. Cytokinesis D was not observed in cells with a deficiency of a cell–cell adhesion molecule (*csB*-null cells, data not shown). However, most of those cells were found to be detached from the substratum because this molecule is also related to cell-substratum adhesion. Therefore, we could not validate this model by the current experiments. In the future, we must conduct experiments on other mutants deficient in cell-cell adhesion molecules.

As another plausible model, asymmetric distribution (therefore, a gradient) of surface molecules of the dividing cells may guide the directional migration of a midwife cell using a haptotaxis mechanism. Nonetheless, the cortical-flow model is more likely than this model because the beads also moved toward the cleavage furrow.

Consequently, we concluded that the midwife mechanism is explained by the cortical-flow model but not by the chemotaxis model.

## Figures and Tables

**Figure 1 cells-08-00473-f001:**
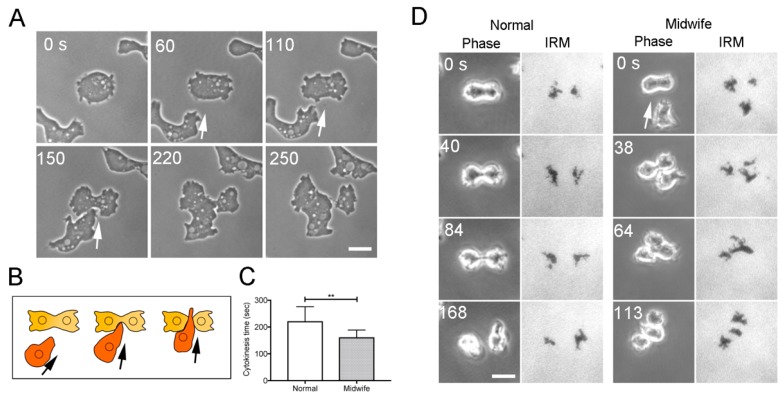
Cytokinesis D depends on migrating neighboring “midwife” cells. (**A**) A representative time course of cytokinesis D according to phase contrast microscopy. The cells were mildly compressed under an agar overlay to improve the image quality. A neighboring cell migrated toward the dividing cell (arrows) and crossed the cleavage furrow. (**B**) A schema to explain cytokinesis D. (**C**) The period from the onset of furrowing to final separation (cytokinesis time) with and without midwife cells. Cells were examined without the agar overlay. Data are presented as the mean ± SD (n > 45, ** *P* ≤ 0.0001, paired *t* test). (**D**) Representative time course data from phase contrast microscopy and IRM of dividing cells without (Normal) cells and with midwife cells (Midwife), respectively. A neighboring cell (arrow) migrated toward a dividing cell and crossed the narrow space between the cleavage furrow and the substratum. Bars, 10 µm.

**Figure 2 cells-08-00473-f002:**
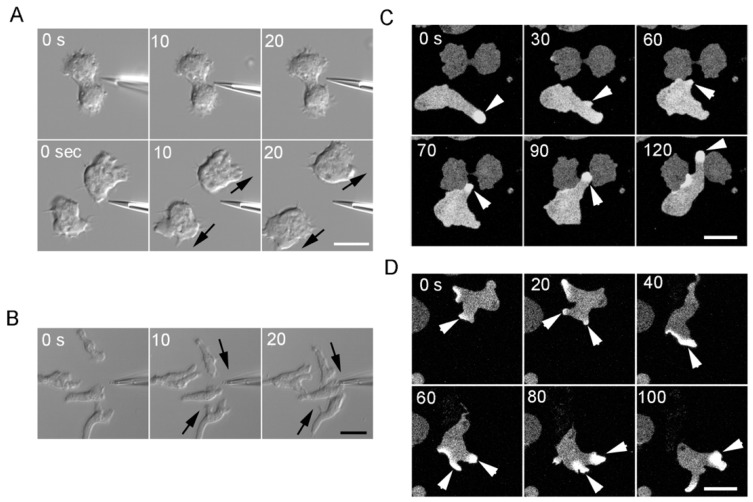
Midwife cells do not respond to the fluid collected near a dividing cell. (**A**) When fluid from the vicinity of the cleavage furrow of a dividing cell was aspirated with a micropipette (upper panels) and discharged near interphase cells (lower panels), the interphase cells neither extended a directed pseudopod nor followed the micropipette (n = 51). (**B**) As a control, when a micropipette containing 10 µM cAMP was applied to developed cells, they migrated toward the micropipette. (**C**) The GFP–PH domain (arrowheads) localized at the leading edge of migrating midwife cells toward a dividing cell. (**D**) The GFP–PH domain (arrowheads) localized at the leading edge of freely migrating vegetative cells. Bars, 10 µm.

**Figure 3 cells-08-00473-f003:**
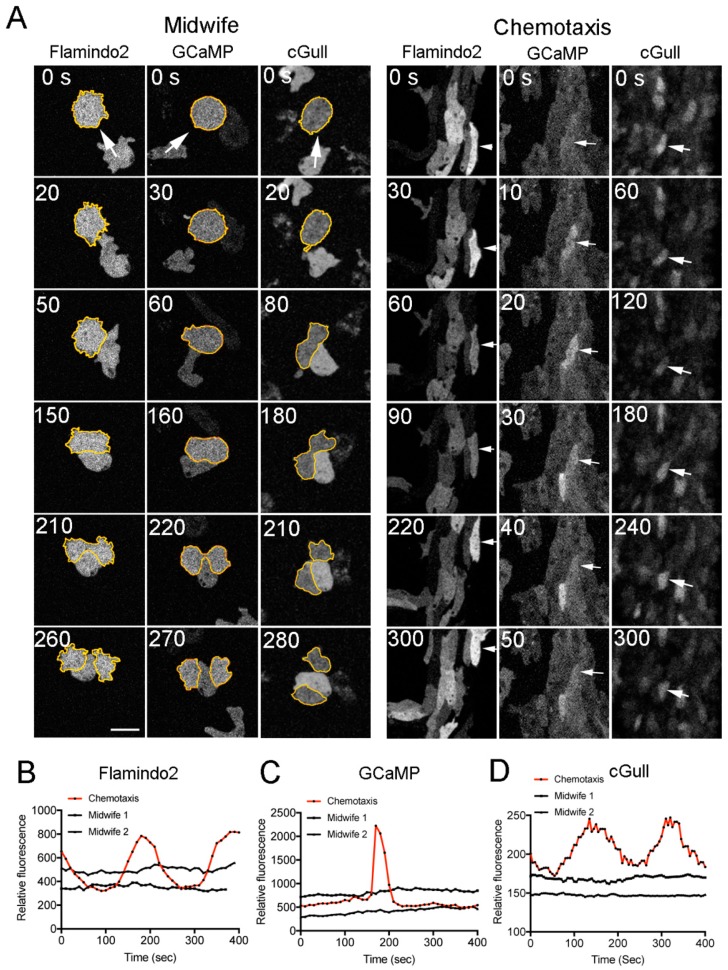
There are no notable changes in Ca_i_^2+^, cAMP_i_, and cGMPi levels in midwife cells. To examine Ca_i_^2+^, cAMP_i_, and cGMP_i_ in midwife cells, Flamindo2, Dd-GCaMP6s, and Dd-Green cGull were expressed in the cells, respectively. (**A**) Midwife cells did not show any notable changes in cAMP_i_, Ca_i_^2+^ and cGMP_i_ levels. Dividing cells are outlined in yellow. As a control, in a cell aggregation stream, the developed cells showed a transient increase in cAMP_i_, Ca_i_^2+^, and cGMP_i_ concentrations (arrows in each panel). Bar, 10 µm. (**B**–**D**) Representative time course data on cAMP_i_, Ca_i_^2+^, and cGMP_i_ in midwife cells (two examples in each graph, black) and a chemotactically developed cell (red), respectively.

**Figure 4 cells-08-00473-f004:**
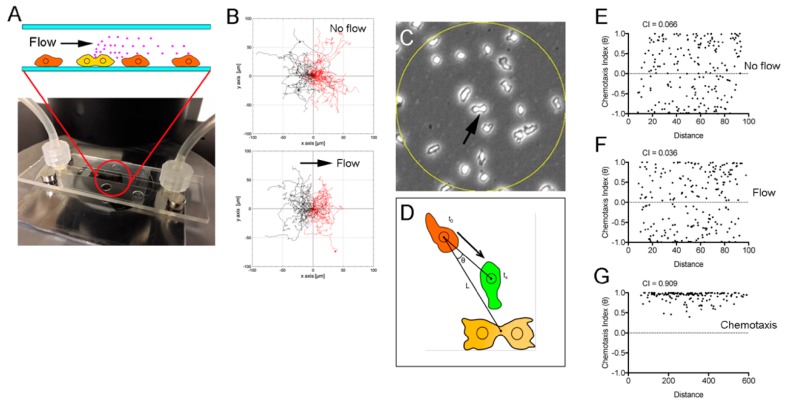
Midwife cells do not migrate toward a dividing cell chemotactically. (**A**) Cytokinesis D was examined in a microfluidic chamber with constant flow. If the medium constantly flows in one direction, a hypothetical unknown substance (red particles) secreted by dividing cells goes with the flow, and midwife cells migrate toward the dividing cell only from downstream locations. (**B**) Tracking of freely migrating cells without flow (upper panel) and with flow (lower panel). Note that the optimized flow did not affect cell migration. (**C**) In the flow chamber, the CI was examined within the circle with a 100-µm diameter (yellow circle) when the dividing cell was located at its center (arrow). (**D**) The CI was defined as the cosine of the angle consisting of three points: the initial position (red), the position of the furrow of the dividing cell (yellow), and the displaced position (green) after 3 min of cell migration. (**E**) Plots of the CI versus the distance of a cell from the dividing cell in the absence of flow (n = 212 cells, three independent experiments). (**F**) Plots of the CI versus the distance of a cell from the dividing cell in the presence of flow (n = 215 cells, three independent experiments). (**G**) Plots of the CI versus the distance of a cell from the aggregation center (n = 157). In this case, the CI was calculated from the angle involving the aggregation center in place of the position of the dividing cell.

**Figure 5 cells-08-00473-f005:**
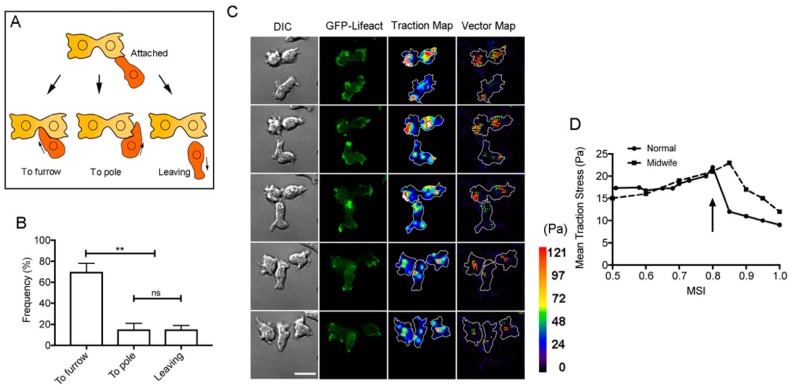
Cell behavior and traction force during cytokinesis D. (**A**) When a migrating cell attaches to a dividing cell, three behaviors were observed. (1) The cell migrates along the surface of the dividing cell toward the cleavage furrow. (2) The cell migrates along the surface of the dividing cell toward the pole. (3) The cell departs from the dividing cell. (**B**) Frequencies of three events. Data are presented as the mean ± SD (n = 62, 43, and 32, three experiments). ** *P* ≤ 0.0001, ns: not significant, *P* > 0.05. (**C**) A typical DIC image, a fluorescence image of GFP–lifeact (a marker of actin filaments), a traction map, and the traction vector map during cytokinesis D. The color code indicates the magnitude of traction stress. Bar, 10 µm. (**D**) Representative time course data on the mean traction stress of a dividing cell with (Normal) and without midwife cells (Midwife). To normalize the cell division stages, the mitosis stage index (MSI) was employed as described in Materials and Methods. The arrow indicates the timing when the midwife cell passed through the cleavage furrow of a dividing cell.

**Figure 6 cells-08-00473-f006:**
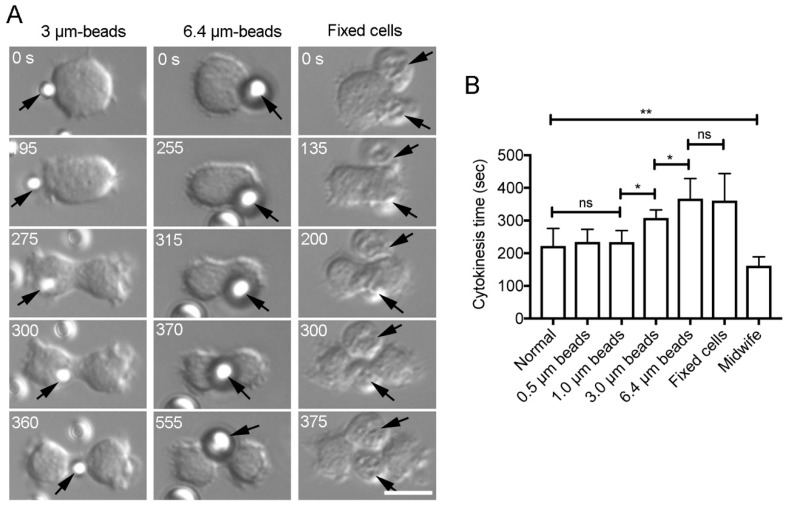
The cortical flow of attached beads. (**A**) Representative time course data on the movement of beads (3 µm and 6.4 µm in diameter) and fixed cells attaching to dividing cells. Note that the beads and the fixed cells (arrows) move toward the cleavage furrow. Bar, 10 µm. (**B**) Cytokinesis time when the beads or fixed cells were attached to dividing cells. Data are presented as mean ± SD (n > 40, three experiments). * *P* ≤ 0.001. ** *P* ≤ 0.0001. ns: not significant, *P* > 0.05.

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
