# Peer review of "Cytokinesis D is Mediated by Cortical Flow of Dividing Cells Instead of Chemotaxis"

_cells, 2019, doi:10.3390/cells8050473_

Round 1
Reviewer 1 Report
The paper developed by Tanaka et al. demonstrated that the midwife mechanism of Dictyostelium is not supported by the chemotaxis model but by a proposed cortical flow model. The paper was well-written and the data is well-presented. I agree with the authors that the midwife cell is not attracted by the soluble chemicals released by the dividing cells. Instead, the midwife cells migrate randomly and are guided toward the cleavage furrow area by the cortical flow after contacting the dividing cells. I agree that the cortical flow may contribute to guiding the midwife cell crossing the furrow in a passive manner. Other active factors from the midway cells may play roles as well. For instance, the asymmetric distribution of surface molecules of the dividing cells may guide the directional migration of midwife cell using a haptotaxis mechanism. I suggest the authors adding some comments on these other factors in the discussion. Overall, this is valuable research and I suggest accepting this paper.
Author Response
Reviewer 1
Comment: I agree that the cortical flow may contribute to guiding the midwife cell crossing the furrow in a passive manner. Other active factors from the midway cells may play roles as well. For instance, the asymmetric distribution of surface molecules of the dividing cells may guide the directional migration of midwife cell using a haptotaxis mechanism. I suggest the authors adding some comments on these other factors in the discussion.
Answer: The reviewer’s comment is very helpful. However, haptotaxis model is unlikely because the beads also moved toward the cleavage furrow. We would like to include the haptotaxis mechanism in discussion as follows.
Page 9:As another plausible model, asymmetric distribution (therefore, a gradient) of surface molecules of the dividing cells may guide the directional migration of a midwife cell via a haptotaxis mechanism. Nonetheless, the cortical-flow model is more likely than this model because the beads also moved toward the cleavage furrow.
Reviewer 2 Report
In this very interesting manuscript Tanaka et al. present a revised concept for cytokinesis D in Dictyostelium. Previously it has been postulated that cytokinesis D is mediated by chemotaxis of neighboring cells towards dividing cells. However, here the authors convincingly show that cytokinesis D is not mediated via chemotaxis and based on their results they propose alternatively that randomly moving cells by change attached to dividing cells and end up at the furrow by cortical flow.
Concerns:
1. In many experiments the authors use cells moving/responding towards cAMP as a control. However, this is comparing vegetative dividing vs moving starved cells. Why did the authors not use vegetative cells chemotaxing towards folate as a control? I don’t expect the authors to repeat all these experiments and it will not change the major conclusions of the paper, but in my opinion it would be a more appropriate control.
2. Section 3.3 verification of the chemotaxis model: The authors showed that the intracellular levels of cAMP, Calcium and cGMP are not changing. However, since the putative chemotactic substance is unknown, based on what is it expected that these levels have to change? The alternative option is that the cells are responding to the putative chemotactic substance, however that these pathways do not involve cAMP, Calcium and cGMP. Please mention in the revised manuscript.
3. Dictyostelium chemotaxis to most attractants is mediated via G-protein coupled receptors. Therefore, Gbeta is crucial for chemotaxis: can Gbeta-null cells divide via cytokinesis D?
Author Response
Reviewer 2
Comment 1:In many experiments the authors use cells moving/responding towards cAMP as a control. However, this is comparing vegetative dividing vs moving starved cells. Why did the authors not use vegetative cells chemotaxing towards folate as a control? I don’t expect the authors to repeat all these experiments and it will not change the major conclusions of the paper, but in my opinion it would be a more appropriate control.
Answer: Folic acid is known as a chemoattractant for vegetative cells. However, axenic AX2 strain used in this paper showed very weak chemotaxis as previously described by Veltman et al. (J Cell BIol., 204,497-505, 2014).
We would like to insert a following sentence;
Page 5: Incidentally, we applied folic acid, a chemoattractant for vegetative cells, to vegetative AX2 cells, but the axenic AX2 strain used in this study showed little or no response; this finding is consistent with another report [14]. Therefore, we applied cAMP to developed cells as a control.
Comment 2. Section 3.3 verification of the chemotaxis model: The authors showed that the intracellular levels of cAMP, Calcium and cGMP are not changing. However, since the putative chemotactic substance is unknown, based on what is it expected that these levels have to change? The alternative option is that the cells are responding to the putative chemotactic substance, however that these pathways do not involve cAMP, Calcium and cGMP. Please mention in the revised manuscript.
Answer: We would like to add a following sentence+
Page 5: On the other hand, the putative chemotactic substance may not necessarily affect cAMPi, Cai2+, and cGMPi.
Comment 3. Dictyostelium chemotaxis to most attractants is mediated via G-protein coupled receptors. Therefore, Gbeta is crucial for chemotaxis: can Gbeta-null cells divide via cytokinesis D?
Answer: Nagasaki et al. (2008) showed that Gbeta-null cells showed a lower frequency of cytokinesis D than wild type cells. We speculate that the reason of the lower frequency is that Gbeta-null cells migrate much slower than wild type cells, which reduces the frequency of attachment or collision of midwife cells to dividing cells.
We would like to add a following sentence.
Page 5:Nagasaki and Uyeda [6]have reported that mutant cells deficient in the G protein bsubunit, which mediates Dictyosteliumchemotaxis via surface receptors, show a lower frequency of cytokinesis D than do wild-type cells. We can explain this finding as follows: most of mutant cells cannot encounter dividing cells in time because the average velocity of mutant cells (1.96 ± 0.57 µm/min, n = 60) is much lower than that of wild-type cells (5.67 ± 1.28 µm/min, n = 60).
Reviewer 3 Report
This is an interesting study on cytokinesis in Dictyostelium. I only have a few minor comments:
1. in figure 2A and B, since cytokinesis was observed during vegetative stage, the control should be folate applied to undeveloped cell (vegetative cell), not cAMP;
2. in figure 3BCD, cAMP induced chemotactic signaling responses supposed to be fast, usually within 10s; however, the response peaks appear around 200s; is this unexpected or inconsistent with other report?
3. in the same figure, how many cells were used to get the result? can they show the graph with statistical analysis?
Author Response
Reviewer 3
Comment 1. in figure 2A and B, since cytokinesis was observed during vegetative stage, the control should be folate applied to undeveloped cell (vegetative cell), not cAMP.
Answer: Folic acid is known as a chemoattractant for vegetative cells. However, Axenic Ax2 strain used in this paper show a very weak chemotaxis as previously described by Veltman et al. (J Cell BIol., 204,497-505, 2014).
We would like to insert a following sentence;
Page 5: Incidentally, we applied folic acid, a chemoattractant for vegetative cells, to vegetative AX2 cells, but the axenic AX2 strain used in this study showed little or no response; this finding is consistent with another report [14]. Therefore, we applied cAMP to developed cells as a control.
Comment 2. in figure 3BCD, cAMP induced chemotactic signaling responses supposed to be fast, usually within 10s; however, the response peaks appear around 200s; is this unexpected or inconsistent with other report?
Answer: Regarding cAMPi and Cai2+response, the observed response was similar to those of previous reports cited in the text. The duration of response in cAMPi depends on the developmental stage as previously described (Hashimura et al., 2019). The cGMPiwas previously measured by using isotope. In the present study, for the first time, we showed the responses of cGMPiby live imaging. The response was much longer than the previous observation as the reviewer pointed out. However, this observation is in the aggregation stream. On the other hand, the previous experiments were performed when cAMP was artificially applied to suspended cells, which may cause the apparent discrepancy. We would like to explain the details in a future paper.
Comment 3. in the same figure, how many cells were used to get the result? can they show the graph with statistical analysis?
Answer: We would like to add the number of observed cells. We would like to keep the accurate and statistical data on the timing of the cAMP-induced cGMPiresponses in the future paper.
Page 5: Similar results were confirmed in multiple cells (n > 30) in each case.
Reviewer 4 Report
With their manuscript Tanaka and co-workers address the long-lasting question of how dividing amoeboid cells are assisted in cytokinesis by so-called midwife cells. According to the current accepted hypothesis this so-called cytokinesis D is driven by a chemotactic signal of the dividing cells, which is received by the midwife cell. Among other data, this hypothesis was based on the observations that midwifery did not occur in Dictyostelium cells lacking a subunit of heterotrimeric G-proteins and that midwife cells show enrichment of PIP3 at the leading edge. With convincing figures Tanaka and co-authors present compelling evidence that this hypothesis has to be refuted, i.e. that midwifery does not involve transmission of a chemotactic signal. Instead they suggest that the leading edge of midwife cells are guided by actin flow towards the constriction site once midwife cells have established physical contact to the dividing cells. Due to the importance and novelty of these findings this manuscript certainly deserves publication in “Cells”. I have only a few critical remarks that should be addressed in a revised version of this manuscript and that do not require further experimentation.
The interesting conclusions of this paper would deserve a more informative title.
please include video data for the individual figures as supplemental material.
p. 1, line32: the category “protozoa” should not be used here as it is very clear that protozoa are no monophyletic group. I strongly prefer to talk about “amoebozoa” in this context.
p. 5, Fig. 3: I miss a number for the sample size (n = ?)
p. 5, line 188: microfluidic instead of microfluid
Fig. 5D: explain the abbreviation MSI
p. 8, line 257 and following: I would like to read some discussion about the question what could transmit the actin flow, which takes place at the cell cortex in the cytosol to the outer face of the plasma membrane, where the midwife interacts. I could also imagine that a gradient of a Rho-GTPase is involved here. In mammalian cells microtubule ends activate RhoA in the area of the cleavage furrow through centralspindlin, which in parallel antagonizes with Rac1. Thus, here there is a reciprocal gradient of these two Rho-GTPases along the plasma membrane. A role of microtubule ends in cleavage furrow formation has also been shown in Dictyostelium.
In the context of the new model, are there ideas why midwifery is dependent on intact heterotrimeric G-proteins, as reported by Nagasaki et al.?

Author Response
Reviewer 4
Comment 1: The interesting conclusions of this paper would deserve a more informative title.
We would like to change the title as follows.
Cytokinesis D is mediated by cortical flow of dividing cells but not chemotaxis
Comment 2: Please, include video data for the individual figures as supplemental material.
Answer: We added videos as supplemental materials for several important figures.
Comment 3: p. 1, line32: the category “protozoa” should not be used here as it is very clear that protozoa are no monophyletic group. I strongly prefer to talk about “amoebozoa” in this context.
Answer: We would like to exchange the term “protozoa” to “amoebozoa”, as the reviewer suggested.
Comment 4: p.5, Fig. 3: I miss a number for the sample size (n = ?)
Answer: We would like to add the sample size.
Page 5: Similar results were confirmed in multiple cells (n > 30) in each case.
Comment 5: p.5, line 188: microfluidic instead of microfluid
Answer: We would like to change the term to ‘microfluidic’.
Comment 6: Fig.5D: explain the abbreviation MSI
Answer: MSI is defined in Materials and Methods. However, we would like to add a following explanation in the legend.
Legend in Fig. 5: To normalize the cell division stages, the mitosis stage index (MSI) was employed as described in Materials and Methods.
Comment 7: p.8, line 257 and following: I would like to read some discussion about the question what could transmit the actin flow, which takes place at the cell cortex in the cytosol to the outer face of the plasma membrane, where the midwife interacts. I could also imagine that a gradient of a Rho-GTPase is involved here. In mammalian cells microtubule ends activate RhoA in the area of the cleavage furrow through centralspindlin, which in parallel antagonizes with Rac1. Thus, here there is a reciprocal gradient of these two Rho-GTPases along the plasma membrane. A role of microtubule ends in cleavage furrow formation has also been shown in Dictyostelium.
Answer: The question is how the cortical flow is regulated, which is very difficult to be explained at present. Actually, Dictyostelium cells have neither RhoA nor centralspindlin. We already have some evidence of the role of microtubule ends, but we would like to keep them in future paper.
Comment 8: In the context of the new model, are there ideas why midwifery is dependent on intact heterotrimeric G-proteins, as reported by Nagasaki et al.?
Answer: We would like to add a following sentence;
Page 5:Nagasaki and Uyeda [6]have reported that mutant cells deficient in the G protein bsubunit, which mediates Dictyosteliumchemotaxis via surface receptors, show a lower frequency of cytokinesis D than do wild-type cells. We can explain this finding as follows: most of mutant cells cannot encounter dividing cells in time because the average velocity of mutant cells (1.96 ± 0.57 µm/min, n = 60) is much lower than that of wild-type cells (5.67 ± 1.28 µm/min, n = 60).